# Bias Correction of Long-Path $CO_2$ Observations in a Complex Urban Environment for Carbon Cycle Model Inter-Comparison and Data Assimilation

T. Scott Zaccheo[1], Nathan Blume[2], Timothy Pernini[1], Jeremy Dobler[2], Jinghui Lian[3]

[1]Atmospheric and Environmental Research, Inc., Lexington, Massachusetts, 02421, USA
[2]Spectral Sensor Solutions LLC, Fort Wayne, Indiana, 46818, USA
[3]Laboratoire des Sciences du Climat et de l'Environnement, CEA-CNRS-UVSQ, Université Paris-Saclay, Gif-sur-Yvette, FR

*Correspondence to*: T. Scott Zaccheo (szaccheo@aer.com)

**Abstract.** The Greenhouse gas Laser Imaging Tomography Experiment (GreenLITE™) trace gas measurement system, jointly designed and developed by Atmospheric and Environmental Research, Inc. and Spectral Sensor Solutions LLC, provides high-precision, long-path measurements of atmospheric trace gases including $CO_2$ and $CH_4$ over extended (0.04 km$^2$ – 25 km$^2$) areas of interest. In 2015, a prototype unit was deployed in Paris, France to demonstrate its ability to provide continuous observations of $CO_2$ concentrations along horizontal air segments and two-dimensional (2-D) maps of time-varying $CO_2$ concentrations over a complex urban environment. Subsequently, these data have been adapted to create a physically consistent set of horizontal segment mean concentrations for: 1) Comparisons to highly accurate in situ point measurements obtained for coincident times within the Greater Paris area, 2) Inter-comparisons with results from high-spatial and temporal regional carbon cycle model data, and 3) Potential assimilation of these data to constrain and inform regional carbon cycle modeling frameworks. To achieve these ends, the GreenLITE™ data are calibrated against precise in situ point measurements to reconcile constant systematic as well as slowly varying temporal differences that exist between in situ and GreenLITE™ measurements to provide unbiased comparisons, and the potential for long-term co-assimilation of both measurements into urban-scale emission models. While both the constant systematic biases and the slowly varying differences may have different impacts on the measurement accuracy and/or precisions, they are in part due to a number of potential common terms that include limitation in the instrument design, uncertainties in spectroscopy and imprecise knowledge of the atmospheric state. This work provides a brief overview of the system design and the current gas concentration retrieval and 2-D reconstruction approaches, a description of the bias correction approach, the results as applied to data collected in Paris, France, and an analysis of the inter-comparison between collocated in situ measurements and GreenLITE™ observations.

## 1. Introduction

GreenLITE™, jointly designed and developed by Atmospheric and Environmental Research, Inc. and Spectral Sensor Solutions LLC, provides high-precision, long-path measurements of atmospheric trace gases including $CO_2$ and $CH_4$ over

extended (0.04 km$^2$ – 25 km$^2$) areas of interest. The system was designed to provide integrated horizontal segment (chord) measurements that intersect the overall region of interest and compliment traditional high-precision in situ point source measurements, and to combine these column measurements with sparse tomography applications that provide 2-D representations of time-varying concentrations and potentially fluxes. GreenLITE™ comprises: 1) a hardware element

consisting of Intensity Modulated Continuous Wave (IMCW) differential absorption spectroscopy transceivers, retroreflectors, data collection/transmission electronics, and a suite of local in situ weather sensors, and 2) a computational element consisting of on- or off-premises gas concentration retrieval, 2-D reconstruction, and web-based data distribution elements. Both hardware and computational elements are outlined briefly below and described in Dobler et al. (2017). The initial purpose behind the system deployment in Paris, France, in 2015 was to demonstrate the ability to provide time-varying average urban-

scale measurements of $CO_2$ emissions and/or consumption on a city sector scale. In addition, these data were used to show the potential for combining these integrated horizontal concentrations with sparse tomography methods to construct 2-D estimates of city-scale greenhouse gas (GHG) variability and dynamics on appropriate spatial and temporal scales. One of the long-term issues with long-path measurements on the order of hundreds to thousands of meters of open-air trace gas concentrations is in providing independent validation of their error characteristics. Unlike point source measurements that can

be calibrated using synthetic input gas with known concentrations traceable to an established standard, these technologies have no well-established mechanism for cross-comparison to a traceable standard reference. In the interim, in order to effectively compare these data with other quasi-collocated high-precision in situ measurements, to incorporate the retrieved long-path concentration values into standardized GHG flux modeling frameworks (Broquet et al., 2011; Göckede et al., 2010; Nehrkorn et al., 2010), or to provide meaningful 2-D maps of concentrations or flux distributions over the field of view, quasi-stationary

biases between the different measurement types (e.g. those from established in situ instruments and a long-path differential absorption spectrometer like GreenLITE™) must be addressed. The observed slowing varying biases or differences between the GreenLITE™ and in situ measurements may be attributed to several sources. Some differences may be attributed to specifics such as lack of automated closed-loop adjustment of wavelengths (use of a standard gas cell to provide absolute wavelength stabilization) in the prototype instrument design, while others are related to the broad category of measurement

techniques that employ spectroscopic knowledge with ancillary measurements of the current atmospheric state to derive estimates of column concentrations. While it is unlikely that one can derive the contributions from each of these potential error terms precisely, a method has been developed, based on physically consistent constraints, to reconcile the overall quasi-stationary differences between local in situ measurements and average GreenLITE™ observation values. This enables the comparison between the two collocated measurement types and the ability to include these data into regional flux models,

while preserving variations or differences due to local sources.

In this work we provide an overview of the GreenLITE™ measurement technology, a description and scientific basis for the proposed bias correction method, a description of in situ point measurement data collected in conjunction with Pierre and Marie Curie University (UPMC, now Sorbonne University) and the Laboratoire des Sciences du Climat et de L'Environnement (LSCE) over a one-year period in Paris, France, a comparison of collocated in situ point measurements and bias-corrected

GreenLITE™ measurements, and analysis of the provided results. The bias-correction process described in this work, provides a mechanism for reducing long-term/multi-day time varying differences between the GreenLITE™ path and in situ measurements that vary in the range of ±10ppm prior to the correction process to near zero (< 0.5ppm).

## 2. Methodology

The GreenLITE™ instrument is a unique implementation of an IMCW Laser Absorption Spectroscopy (LAS) method that was initially developed as an airborne demonstrator for spaceborne application (Dobbs et al., 2007; Dobler et al., 2013). The GreenLITE™ system utilizes two transceiver units and a series of corner cube reflectors to generate a series of overlapping atmospheric differential transmission measurements (see Figure 1 for an example). Each transceiver utilizes a pair of fiber coupled semiconductor lasers, with wavelengths selected for the specific gas being probed such that one laser wavelength is

absorbed while the other experiences less to no absorption. Typically, the desired wavelengths are only a few 10's of pm apart making them difficult to separate optically, while one can use time division through sending pulses of each individual wavelength and collecting them one at a time. Several noise sources such as scintillation make this approach challenging. For the IMCW approach each wavelength is uniquely modulated in amplitude and combined in fiber before transmission from the coaxial

telescope, the light travels the same path to the reflector, and upon return to the telescope is collected simultaneously and converted to an electrical signal with a photodetector. This approach allows for simultaneous transmission and reception of multiple wavelengths where the individual wavelength light intensities can be distinguished in the digital domain

through the use of a lock-in amplifier or matched filter. Since the differential transmission is determined through a ratio of the simultaneously transmitted and received signals at the different wavelengths, there are a number of noise terms that are common in the IMCW approach, due to simultaneous transmission of the online and

offline wavelengths, that cancel out but would remain independent in traditional pulsed method. The system is designed such that the telescope is mounted on a mechanical scanner and is fiber coupled to a stationary electronics chassis which houses the lasers, modulators, detectors, computer, and other associated electronics. Both the optical head and the

electronics box are temperature controlled and have been deployed remotely as both portable configurations and semi-permanent installations (Dobler et al., 2017). The installation in Paris was of the semi-permanent design and consisted of steel tubing

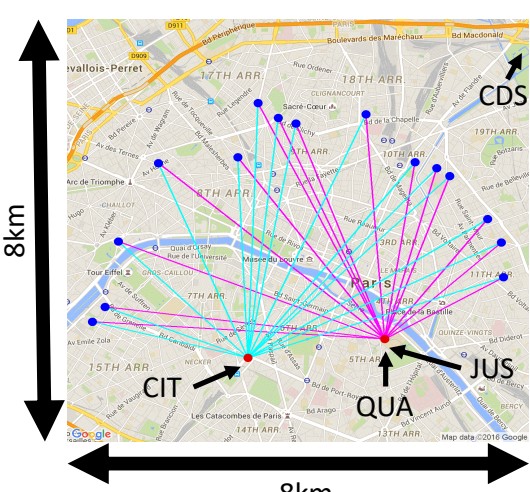

**Figure 1. Example GreenLITE™ system layout in Paris, France. The red dots in the images denote the transceiver locations (CIT and Jussieu [JUS]), and the blue dots indicate the placement of 15 retroreflectors. The cyan and magenta lines depict the lines of sight between each transceiver and the retroreflectors. The locations of the two in situ are also indicated by the two black arrows, Cite des Sciences et de l'Industrie (CDS) and QUALAIR laboratory (QUA). QUA is located in close proximity to the Jussieu tower on the same campus.**

frames secured by concrete blocks and tethers for safety. One transceiver was installed on the top of the Jussieu tower at UPMC, while the other transceiver was mounted on the roof of the Tour CIT Montparnasse building (CIT). The locations were selected to put the two transceivers ~2.5 km apart while maintaining similar heights above sea level. A total of 15 reflectors were installed on various buildings with the requirement that they were < 5.5 km from each transceiver. In addition to the installation of two transceivers and 15 retroreflectors, a Davis weather station was also installed at each transceiver location as an additional input to the atmospheric state parameters.

The GreenLITE™ system deployed to Paris used a custom 15.25 cm F/2 receiver telescope with a 2.54 cm transmitter through a hole in the center of the primary receiver. Approximately 25 mW of optical power was used for the combined on- (1571.112 nm) and off- (1571.061 nm) line fiber coupled distributed feedback laser channels, keeping the system well below eye-safe limits. The modulation frequencies were near 50 kHz, and the data from each reflector was integrated for ten seconds resulting in a measurement of all chords, including calibration and slew time, about every four minutes. An image of a GreenLITE™ transceiver deployed in Paris is shown in Figure 2. The data from the systems were transmitted continuously via a 4G wireless network connection to the cloud-based data storage and processing sub-system described below.

GreenLITE™ instrument samples collected over the period between November 2015 and November 2016 were processed using a multi-threaded cloud-based product generation sub-system that converts the observed differential optical depth values to $CO_2$ column densities and combines them to construct 2-D maps of $CO_2$ concentrations in near-real time. This process, illustrated in Figure 3 and described in detail in Dobler et al. (2017), continuously computes integrated column $CO_2$ concentrations along horizontal lines of sight between the transceivers and reflectors and estimates the 2-D distribution of time-varying $CO_2$ concentrations based on an analytic model and an ensemble set of temporally concurrent integrated horizontal path or "chord" measurements. The first step in this process converts each GreenLITE™ transceiver's differential transmission measurement to an

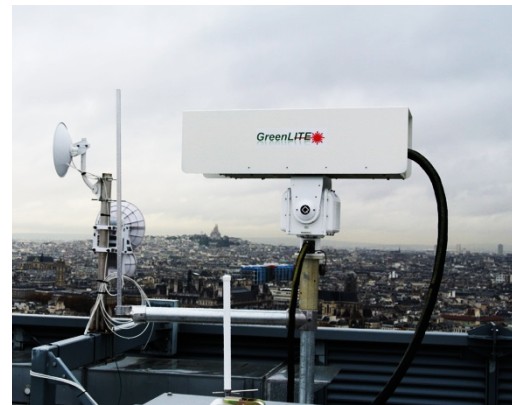

**Figure 2. Image of a GreenLITE™ transceiver atop the Jussieu tower. The pair of transceivers were deployed in Paris, France from November 2015 through November 2016.**

estimate of the horizontal column $CO_2$ concentration in parts per million volume (ppmv). It combines the difference in observed "on" and "off" line optical depth values ($\Delta\tau = \tau_{on} - \tau_{off}$), measured meteorological state (air temperature (T), air pressure (P), and relative humidity (RH)), and static path length parameters with a radiative transfer (RT) based iterative retrieval scheme that minimizes the difference between the observed and modeled differential optical depths given coincident measurements of the atmospheric state. This RT approach employs a line-by-line model, LBLRTM (Clough et al., 2005), in conjunction with a standard steepest descent search technique to provide a model value of $\Delta\tau$ that best matches the observed $\Delta\tau$ given the atmospheric state along the chord.

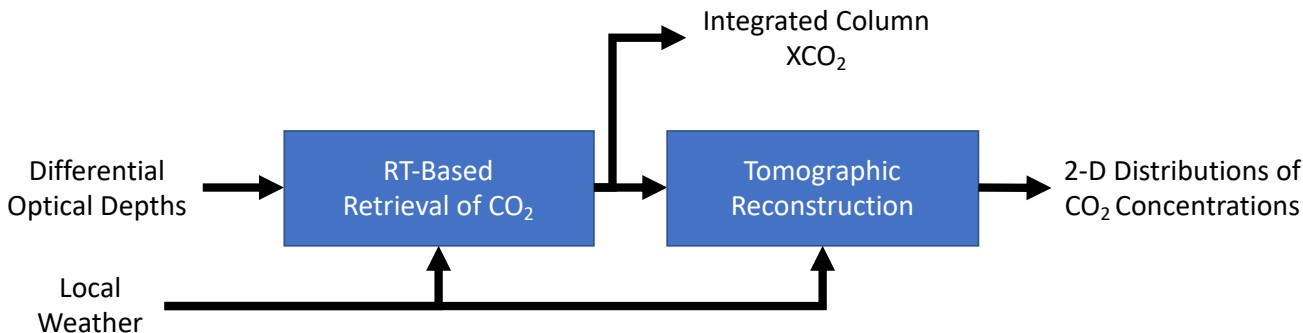

**Figure 3.** **Block diagram of the GreenLITE™ remote data processing stream from differential optical depths to 2-D distributions of CO₂ concentrations.**

In short, the RT model, given the measured atmospheric state and an initial guess at the column concentration, is used to compute a modeled differential optical depth

$$\Delta\tau_M = \tau_{Mon} - \tau_{Moff} \tag{1}$$

where $\tau_{Mon}$ is the model optical depth at the monochromatic online wavelength, and $\tau_{Moff}$ is that at the offline wavelength. Next, the observed $\Delta\tau$ is compared to the model value ($\Delta\tau_M$), and the resulting difference is used to update the estimated column concentration. This is achieved using the gradient or finite differences defined by the change in column concentration as a function of the change in optical depth. In this work, the gradient is defined as

$$\nabla CO_2 = \frac{\Delta CO_2}{\Delta\tau_M - \Delta\tau_{M+1}} \tag{2}$$

where $\Delta\tau_{M+1}$ is the modeled differential optical depth at an enhanced column CO₂ value which is the last estimated CO₂ value plus a constant offset $\Delta CO_2$. $\Delta CO_2$ was set to a fixed value of +2 ppmv. Finally, an updated estimate of the column concentration is computed based on

$$CO_{2(n+1)} = CO_{2(n)} + \nabla CO_2 (\Delta\tau_M - \Delta\tau) \tag{3}$$

where $CO_{2(n)}$ represents the previously estimated column concentration value, and $CO_{2(n+1)}$ is the updated value. This iterative process is continued until the absolute difference, $|\Delta\tau_M - \Delta\tau|$, falls below the instrument noise threshold or the number of iterations exceeds a user-defined maximum. Nominally, this method converges in one to two iterations under a wide variety of atmospheric and environmental conditions.

A multi-segment approach was employed as part of this process to minimize impacts of gradients in T/RH/P along slanted paths with lengths greater than 1 km. In these cases, a chord (observational line of sight) was divided into multiple segments, each segment was assigned different estimates of T/RH/P using a distance-weighted average of the surrounding weather stations, and a value of P was computed on a per chord basis using a lapse rate equation to account for changes in height above sea level at each chord segment. The weather data was obtained from weather stations located at each transceiver, and others,

owned and operated by local institutions or individuals, scattered throughout the area of interest. Prior to ingesting these data, the provided temperature, %RH and surface pressure from each was evaluated against local standard for quality assurance and control. Finally, the $CO_2$ concentration of each chord segment is retrieved and the final estimate of $CO_2$ concentration along the full path length is computed as a weighted sum of segment values.

The resulting retrieved GreenLITE™ concentration values were initially employed in an additional minimization scheme that constructs real-time 2-D views of the underlying field concentrations based on sparse tomography methods. While traditional tomographic applications employ large numbers of back projections (chords) and angles that approach or exceed the number of pixel elements in the resulting 2-D image, this application is under-sampled due to the number of deployed transceivers and reflectors, site topography, and both natural and man-made barriers. This approach, detailed in Dobler et al. (2017), is similar

to those proposed and implemented in Levine et al. (2016), Cuccoli et al. (2009), and Giuli et al. (1991, 1999). The data were archived for post-deployment analysis. Over the period between November 22, 2015, and November 16, 2016, more than 1.4M collocated GreenLITE™ measurements of differential optical depth and in situ measurements of near-surface temperature, moisture, atmospheric pressure, wind speed, and wind direction were collected and archived.

### 2.1.   Post-Deployment Calibration of Chord Concentration Data

The initial real-time estimates of column concentrations were computed using version 12.2 of LBLRTM, spectroscopy coefficients developed based on HITRAN 2008 (Rothman et al., 2009), and on- and off-line wavelengths that were measured and adjusted periodically over the period of the campaign. Version 12.2 of LBLRTM employs a Voigt line shape function at user-defined atmospheric levels, a model of the continuum that includes self- and foreign-broadened water vapor as well as continua for $CO_2$, $O_2$, $N_2$, $O_3$, and extinction due to Rayleigh scattering. The initial spectroscopy database included updates to

the $CO_2$ line parameters and coupling coefficients derived from the work of Devi et al. (2007a) and Devi et al. (2007b). While every effort was made during deployment to calibrate the system and maintain a stable baseline, no real-time independent data were available for cross-comparison. In addition, this first-generation design showed a slight systematic long-term drift in both the on- and off-line wavelengths as a function of continuous operation of the lasers. The observed drift was partially corrected for by remotely nudging the on-line wavelength to a point of maximum absorption for each transceiver given a fixed

reflector position. Once the maximum absorption value was obtained, it was assumed that the corresponding wavelength was that of the modeled peak absorption given local atmospheric conditions. Finally, the identical adjustment, in terms of digital counts on the thermal controller, was made to the off-line wavelength in an attempt to keep both synchronized with the original measured values, since the offline wavelength does not have the same natural reference point.

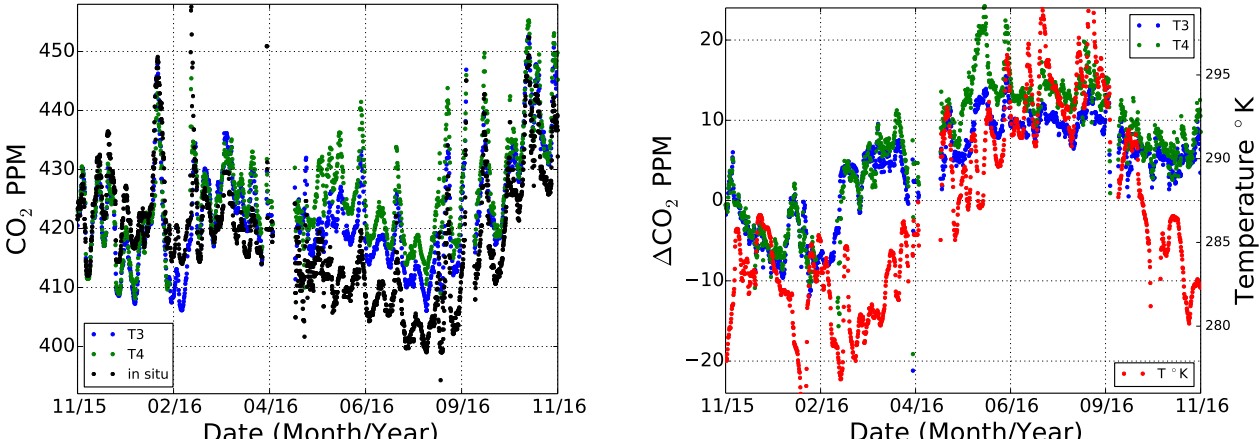

**Figure 4. Four-day median measured and observed CO₂ over all chords for each transceiver. The left-hand figure illustrates the median value for each transceiver and the combined in situ measurement median values. The right-hand plot shows the differences between the median transceiver values and the median in situ measurements. The additional red trace in the right-hand panel denotes the 4-day median near-surface air temperature and provides a visual aid to assess the correlation between the atmospheric state and the differences between column and point source observations. T3 denotes transceiver serial #3 located in the lower-left hand corner of Figure 1 at the CIT tower, and T4 indicates transceiver serial #4 located in the lower-right at UPMC.**

While this approach significantly reduced the observed time-varying trends in the GreenLITE™ data, additional systematic differences were observed when compared to long-term averages (days) of independent quasi-coincident in situ point measurements. Post deployment, two sources of collocated in situ measurements were identified through collaboration with LSCE, which continuously operates multiple sites throughout the Paris metropolitan area as part of its urban GHG monitoring system (Bréon et al., 2015; Xueref-Remy et al., 2016). The two most appropriate sites for cross comparison with GreenLITE™ observations were located at the Cite des Sciences et de l'Industrie (CDS) and the QUALAIR laboratory (QUA) at UPMC. The CDS site is located in a park in the northeast of central Paris (48.896°N, 2.388°E) and is equipped with a cavity ring-down spectroscopy (CRDS) analyzer (Picarro G2401). The air is sampled on the roof of the building at 34 m above ground level (AGL). Variations in CO₂ concentrations at CDS range from 390 to 550 ppm (hourly means) over the observation period of 360 days. The instrument at CDS is calibrated every two months using a combination of three reference gases. All reference gases are calibrated at LSCE with a suite of six NOAA/WMO cylinders (WMO-CO2-X2007 scale) providing a measurement accuracy of 0.02 ppm. The second instrument was a Picarro G2410 CRDS installed at the QUALAIR laboratory (http://qualair.aero.jussieu.fr/qualair.php) adjacent to the Jussieu tower (collocated with the GreenLITE™ transceiver shown in the lower right-hand corner of Figure 1) at a height of 122 m above sea level (ASL) (25 m AGL). This instrument is calibrated twice daily with reference gas base in the WMO-CO2-X2008 scale. The long-term time-varying differences between the GreenLITE™ and an average of the two in situ measurements are illustrated in Figure 4.

This figure shows the complex nature of the differences between horizontal chord observations of CO₂ column amounts and the in situ point source measurements. These differences show several similar trends across the two independent transceivers, located 2.4 km apart, but are by no means identical in shape or form and do not seem to be highly correlated with the median

atmospheric state, e.g. near surface air temperature (shown in red). However, several other potential factors may contribute to these differences and range from those attributed to the natural variations between an average measurement along a column of air kilometers in length and one at a highly localized point in a complex urban environment, to uncertainties in spectroscopic and instrument effects. The observed systematic differences must be addressed to provide either meaningful comparisons

between the two sets of observations or to incorporate both sets of data into common analyses and/or models that aid in the development of regional estimates of localized fluxes. While not desirable, it is often necessary to apply post-calibration corrections to such data to rectify residual differences between observation types. An example of such is the use of a scale factor of the form $(1 \pm \varepsilon)$, where $\varepsilon$ is $<< 1$, in the processing of data from the Total Carbon Column Observing Network (TCCON) to rectify differences between observed trace gas column densities and aircraft observations (Wunch et al., 2010).

However, as illustrated in Figure 4, a simple linear combination of small offset and scale factor would do little to minimize the complex differences between the observed GreenLITE™ column values and coincident in situ observations. An alternative approach that is similar in principle and provides a non-linear contribution to the model optical depths is to make slight adjustments to either the on- or off-line wavelengths. Adjusting one of the wavelengths provides a mechanism to compensate for uncertainties in spectroscopy, e.g. absolute line-center and line-shape parameters, and imprecise knowledge of the true off-

line position, whose absolute changes over time were hard to assess indirectly in the absence of a pronounced spectral absorption feature. We therefore examine the use of small temporally-varying adjustments to the off-line wavelength as a non-linear mechanism for minimizing the long-term/multi-day average differences between the GreenLITE™ column amounts and corresponding in situ point measurements.

    Changes in the off-line wavelength modulate $\Delta\tau_M$ in Eq. (1) and the estimates of column dry-air $CO_2$ mixing ratio given in Eq.

(3). This updated algorithmic approach, designed to aid in post-processing calibration of retrieved GreenLITE™ $XCO_2$, consists of a three-step process. In the first step, optimized on- and off-line wavelength are computed for a sparse set of data randomly selected on a transceiver-by-transceiver basis. In the second step, an optimized on- and off-line wavelength is computed for each GreenLITE™ sample based on long-term averages of those values computed in the first step. Finally, the complete set of samples is reprocessed given their newly assigned on- and off-line wavelength values.

The initial step that provides an estimate of optimized on- and off-line wavelength for a time varying base was accomplished by the following sub-steps: 1) Randomly select a sparse set of samples from the entire set of all chords on a transceiver-by-transceiver basis. The average number of chords selected at random was approximately four chords per hour per transceiver. 2) Define the corresponding average temperature, relative humidity, and atmospheric pressure for each selected chord and observation time and use these to compute the nominal optical depths over a $\pm1$ cm$^{-1}$ region centered around the nominal

transceiver on-line wavelength given a nominal background concentration in $CO_2$ of 400 ppmv. Locate the wavelength of maximum absorption and assign this value as the optimized on-line wavelength for the transceiver sample. 3) Compute the optimum off-line wavelength value which minimizes the absolute difference between the GreenLITE™ differential optical depth observation and the average observed in situ measurements from both CDS and Jussieu collocated in time with the selected GreenLITE™ observation. A gradient-based approach, identical to that described above for computing $XCO_2$ from

GreenLITE™ differential optical depth values, was used to locate the off-line wavelength that provides the best match to average in situ measurements, given an estimate of T, RH, and P values along the path of interest and an optimized on-line wavelength value.

Once the entire time sequence of optimized on- and off-line wavelengths for the sparse set of observations was obtained, an optimized on- and off-line value was computed for each GreenLITE™ sample based on the associated transceiver and the median on- and off-line values over a $\pm$2-day period centered around each sample time. Finally, the multi-segment retrieval process described above was used to recompute all the GreenLITE™ column amounts for the entire observation period. The 4-day median approach was arrived at in part based on the calculated drift which was on the order of 0.08 pm/day, and the measurement precision of the absolute wavelengths of < 0.5 pm. The median filter process was designed to balance capturing the observed slowing varying changes in wavelengths, which were partially compensated for by weekly adjustments to lasers, and local variations/differences in concentrations between large-spatial averaged columns over large and diverse areas and point in situ measurements. A 4-day window seemed to provide a reasonable balance, and the use of a median instead of a mean value removed observed biases introduced by large transient differences between the GreenLITE™ data and in situ measurements.

The procedure outlined above was implemented using three consecutive versions of LBLRTM: version 12.2 with an augmented $CO_2$ line parameter database, version 12.8 with version 3.6 of the line parameter database, and version 12.8 that included $CO_2$ line parameters from HITRAN 2016 (Gordon et al., 2017). Version 12.2 was used primarily as an internal engineering validation of the algorithm process and to assess the methodological differences between the original retrieved values and reprocessed data. Retrievals based on version 12.8 were denoted as the current baseline, and the alternative set of retrievals performed using line parameters from HITRAN 2016 were constructed to assess changes due to current spectroscopy. The baseline spectroscopy includes $CO_2$ line parameters from HITRAN 2012 (Rothman et al., 2013) with updates from Devi et al. (2007a), as well as weak-band $CO_2$ line parameters in the 5929-6392 $cm^{-1}$ band provided by OCO-2 ABSCO v5.0 (Payne, 2017). The primary objective behind repeating the correction and retrieval process using both RT model parameters derived from HITRAN 2012 and 2016 was to assess whether or not one or the other parameterization provided a significantly better overall fit to the data for the two nominal wavelengths of interest. While the RT model parameterization is continuously being refined by groups worldwide, changes to the database may or may not have a positive impact on the resulting retrievals that rely on observations at two distinct nearly monochromatic wavelengths.

## 3. Results

An example of the raw resulting structure and observed deviations from in situ measurements is provided in Figure 5, which shows a sample sub-set of the original (blue) and reprocessed (green) GreenLITE™ data and average in situ measurements (red). This subset of data clearly illustrates the similarities and differences between the GreenLITE™ data and in situ measurements, the time varying characteristics of the corrections and the impact of the correction process. Both the left

(transceiver #3 chords) and right (transceiver #4 chords) panels in Figure 5 show significant reduction in the long-term average differences between the observations and in situ values, without impacting data variability and chord-by-chord variations. While the majority of the corrected samples in both the left- and right-hand panels have shifted toward the in situ observations based on the computed bias corrections, a limited set of corrected T4 samples seem to represent a set of organized outliers that

exceed the majority of the normal range described by the T3 data on the left-hand panel and the majority of the samples on the right. While these deviations from the quiescent urban background are not explored in this work, they aid in illustrating the ability of GreenLITE™ to accurately describe the urban background in real-time, localized/region variations in GHG concentrations, as well as potential spatially and temporally varying urban hot-spots that may be denoted by the extreme outliers.

While Figure 5 illustrates a localized sample of the data in time, the full results over the one-year deployment of these reprocessing efforts are shown in Figure 6. This figure illustrates the four-day median-filtered chord values sampled at a six-hour cadence and the difference between those values and the corresponding in situ observations. These plots illustrate the reprocessed results that correspond to the data provided in Figure 4. The plot on the left-hand side of Figure 6 shows the median observed and measured concentrations as a function of transceiver and RT model used to construct retrieved values.

The right-hand side illustrates the differences between median retrieved values and median collocated in situ measurements for all model and transceiver combinations. These plots provide a graphic representation of the differences between retrieved and measured values and show dramatic reduction in systematic long-term biases with little change in daily structure. This is also illustrated by the mean bias and standard deviation values given in

Table 1 where all post-processed biases have been significantly reduced and are now within the GreenLITE™ measurement

error of < 1ppmv, and the four-day average standard deviation between observed and point source in situ measurements are similar in magnitude. Finally, Figure 7 provides results from the post-processing analyses that determined the change in off-line values (in picometers) from the pre-set wavelengths required to construct the data shown in Figure 5 and Figure 6. These results show that, while both instruments required similar adjustments in off-line wavelength to compensate for differences between GreenLITE™ observations and collocated in situ measurements, the RT model coefficients play a significant role in

defining the overall differences between remote sensing-based measurements and in situ observations. The values retrieved using LBLRTM version 12.8 and HITRAN 2012 line parameters required that an average of -4.3 pm offset be added to the off-line wavelength to provide a match to the median in situ measurements, and those obtained using LBLRTM version 12.8 and HITRAN 2016 line parameters required the addition of +3.8 pm on average. This by no means indicates that the GreenLITE™ off-line wavelengths were measured incorrectly. It merely indicates that an offset of either -4.3 pm or +3.8 pm

must be added to the observed off-line in order to correctly compensate for the unknown combination of system inaccuracies in the on- and off-line wavelengths, those associated with the RT model implementation and accompanying line parameters, other instrument-oriented systematic error terms, real effects due to spatial sampling differences between GreenLITE™ and the in situ instruments, and those associated with the in situ observation mechanisms themselves. The right-hand side of Figure 7 also illustrates the observed inverse relationship (r = -0.6) between ambient temperature and the difference between model

results (HITRAN 2012 and HITRAN 2016) for each set of observed transceiver measurements. This plot shows both a constant and a temperature dependent change in spectral knowledge as represented by the data in HITRAN 2012 and 2016 at 1571.061 nm. The constant offset, which is nearly identical for both transceivers, is represented by an approximately 8 pm difference in off-line position needed to compute similar to optical depths, and time varying biases that are inversely proportional with temperature. As the average temperature decreases, the difference between computed GreenLITE™ values increase.

**Table 1. Mean bias values for original and post-processed data. These values represent the observed biases over the entire collection period between the GreenLITE™ observations and average in situ measurements. The average in situ measurements were computed from the collection of all available observations at any instance in time. The average post-processing offsets in wavelengths denote the average difference from measured nominal wavelength settings employed in retrieving the corresponding GreenLITE™ values.**

| Analysis | Transceiver | Average CO$_2$ Difference (GreenLITE™ - in situ) | Average CO$_2$ STD (GreenLITE™ - in situ) | Average Wavelength Offset (Nominal - Postprocessed) |
|---|---|---|---|---|
| Initial real-time retrievals | T3 | 4.17 ppmv | 12.12 | – |
| | T4 | 6.86 ppmv | 10.76 | – |
| Post processing HITRAN 2012 | T3 | -0.11 ppmv | 14.20 | -3.83 pm |
| | T4 | 0.55 ppmv | 11.91 | -4.70 pm |
| Post processing HITRAN 2016 | T3 | -0.54 ppmv | 14.05 | 4.09 pm |
| | T4 | 0.41 ppmv | 11.67 | 3.46 pm |

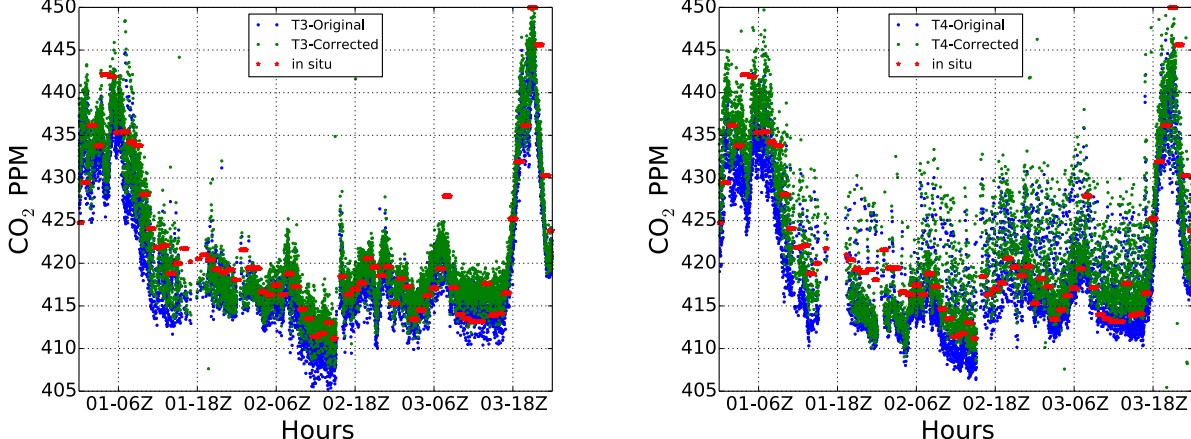

**Figure 5. Sample set of 4-day median post-processed retrieved GreenLITE™ values and corresponding in situ observations for a representative 3-day period in March 2016. The median values were computed every six hours using a 4-day window over all chords for all transceiver/RT-model combinations. The blue data represents the uncorrected results for transceiver 3 (left) and 4 (right), the green data illustrate the reprocessed data using LBLRTM 12.8 and line parameters based on HITRAN 2016 (HT-2016), and the red data shows the average hourly in situ measurements.**

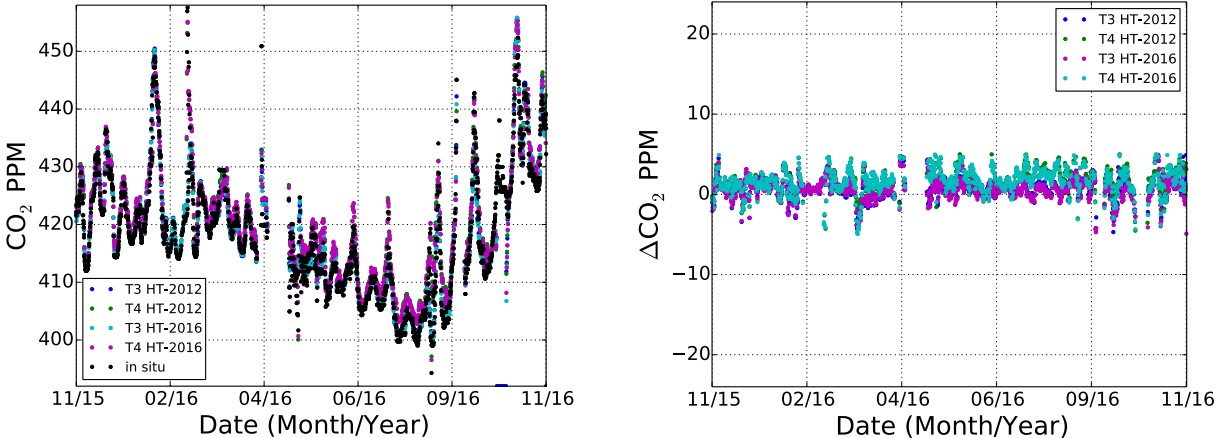

**Figure 6. Four-day median post-processed GreenLITE™ and in situ $CO_2$ values. The median values were computed every six hours given a 4-day window for each transceiver/RT-model combination. The left-hand plot illustrates the median and the combined in situ measurement values, and the right-hand plot shows the differences between the median transceiver values and the median in situ measurements. The blue and green results represent the reprocessing using LBLRTM 12.8 and line parameters based on HITRAN 2012 (HT-2012), and the cyan and magenta denote the post-processed results generated with LBLRTM 12.8 and HITRAN 2016 line parameters (HT-2016).**

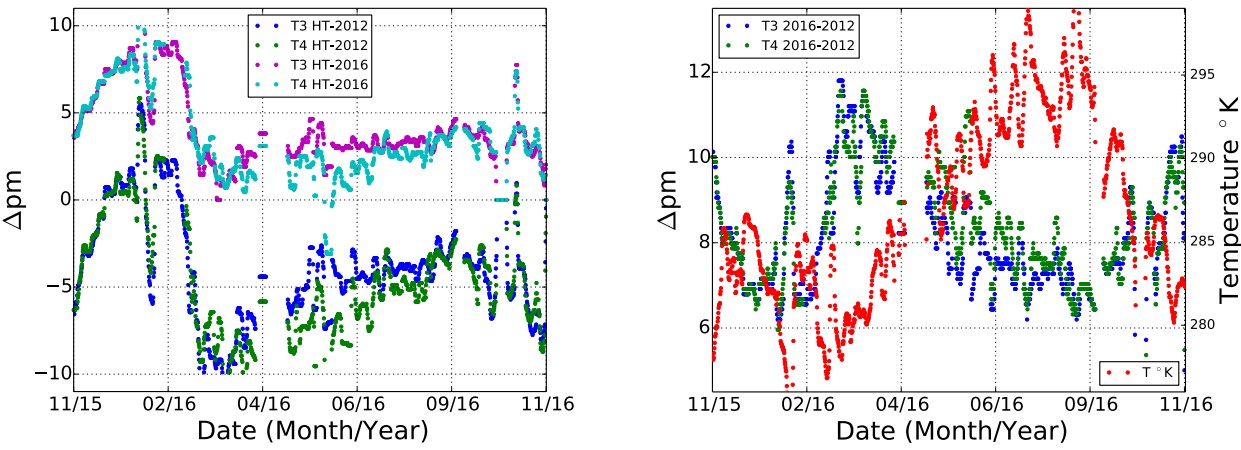

**Figure 7. Four-day median post-processed GreenLITE™ offsets in wavelength from measured or nominally tuned values required to compute column $CO_2$ values given in Figure 6. The median values were computed every six hours over a 4-day window for all chords and each transceiver/RT-model combination. The left-hand plot illustrates the median wavelength offset values, and the right-hand plot shows the differences between the T3 and T4 values along with median air temperature for the same period. The blue and green data represent the wavelength offset for the data reprocessed using LBLRTM 12.8 and line parameters based on HITRAN 2012 (HT-2012), and the cyan and magenta data represent those generated with a combination of LBLRTM 12.8 and line parameters derived from HITRAN 2016 (HT-2016).**

## 4. Conclusions

This work has demonstrated a method for correcting systematic biases in new and novel long-path estimates of GHG concentrations over extended and complex regional domains by co-registering them with precise yet highly localized nearby in situ measurements of GHG concentrations. While the approach presented in this work does not directly address the absolute accuracy of these long-path remote sensing measurements, it does provide a well-defined mechanism for minimizing biases between the long-path measurements and precise in situ measurements that vary slowly in time and are due to a number of subfactors. This work also demonstrates that the defined approach may provide additional mechanisms for minimizing spectroscopic mismatches between observed and modeled long-path differential absorption spectrometer data. In the case illustrated above, retrieved values of integrated GHG concentration based on either HITRAN 2012 and 2016 produce similar to corrected results over a broad range of environmental conditions. While both implementations produce similar corrected results, the required correction factors were significantly different in magnitude, and the difference between the two correction factors remained nearly constant over the period of observation. This constant difference in conjunction with continuous measurements or locking of the on- and off-line wavelengths may provide a metric to assess different RT parameterizations and retrieval approaches.

Rectifying the biases between the two measurement types will enable inclusion of both into complex analysis tools such as regional emission modeling frameworks. Unlike point source measurements that can be bias-corrected and calibrated based on extensive comparisons to known standards in a very small confined space over a wide range of temperatures and pressures, the GreenLITE™ data and other open-path or column observations that span large spaces present new and not yet fully solved absolute calibration challenges. The fundamental challenge is in the construction of a confined long-path environment whose composition along a defined path can be independently verified to compute the error characteristics of the long-path measurements, as well as being systematically varied to represent a wide range of environmental settings. The method described in this work provides an interim step that would enable the meaningful assimilation of both point source and long-path measurements of GHG concentrations into reconstruction approaches or models that provides 2D estimates of time varying trace gas concentration or emission over complex regions of interest.

## 5. Disclaimer

None

## 6. Acknowledgements

We would like to acknowledge our colleagues at Le Laboratoire des Sciences du Climat et de l'Environnement (LSCE), Laboratoire ATmosphères, Milieux, Observations Spatiales (LATMOS), Cité des Sciences et de l'Industrie (CDS) (Marc

Jamous, Jean-Christophe Theisen, Didier Philippe, Michel Maintenant, Sylvain Aulombard, and Michel Pérez), and the staff from LSCE for setting up the analyzer (M. Ramonet, M. Delmotte, and M. de Florinier), for providing support and data from their measurements as comparisons to the GreenLITE™ data, and for continuing with further modeling and comparison activities. Finally, the authors would like to thank François Marie Bréon, Gregoire Broquet and Philippe Ciais of LSCE for their continuing support of the analysis of these data, and their ongoing feedback and insights into the GHG dynamics in the greater Paris area.

## 7. Code/Data Availability

Sample data sets and results corresponding to this study are available upon request from the corresponding author.

## 8. Author Contribution

T. Scott Zaccheo designed, developed and implemented the calibration method presented in this work with contributions and review by all co-authors. Nathan Blume and Jeremy Dobler are the principle architects, implementors and operators of the GreenLITE™ hardware and were responsible for the data collection process. T. Scott Zaccheo and Timothy Pernini co-developed the GreenLITE™ calibration, retrieval and post processing software, and were responsible for the data analysis provided in this work. Jinghui Lian assisted in obtaining and quality controlling the comparison *in situ* measurements from various institutions in the greater Paris area, and provided independent review and assessments of these data. T. Scott Zaccheo prepared the manuscript with contributions from all co-authors.

## 9. Competing Interests

The authors declare that they have no conflict of interest.

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
