# Peer review of "Bias Correction of Long-Path CO2 Observations in a Complex Urban Environment for Carbon Cycle Model Inter-Comparison and Data Assimilation"

_Atmospheric Measurement Techniques, 2019_

## Referee Comment (RC1) · Manuel Queißer (Referee) · 24 Jul 2019

The paper addresses the non-trivial comparison between CO2 concentrations from in situ sensors with those from remote sensors, which measure path-integrated (path averaged) CO2 concentrations. This work represents a very crucial step towards improved quantification of sources and sinks of greenhouse gas emissions as it may provide spatially integrated concentrations improtant for flux retrieval. This is especially important as urban centres become significant producers of greenhouse gases. Regional emission modeling frameworks will benefit from this work.

[Figure]

It is a clearly written paper and the limitations of the approach are discussed in the end so I recommend publication. I just have a few comments, see below.

General comment:

In essence, if I'm not wrong, you calibrate the differential optical depth of your open path differential absorption spectrometer with point optical depths from in-situ sensors. It works and the result is impressive but the method is pragmatic as you state yourselves in the manuscript already. This must assume that the in-situ sensors are accurate, which is reasonable, but only near them, which is the dilemma of comparing path averaged CO2 concentrations with point concentrations. Don't you risk adding a new bias by this method? In that context it would be interesting to know how much of the difference between GreenLite and the in-situ values are caused by instrument bias and random error and how much are actually "real", that is, physical differences in CO2 concentrations due to the different volumes probed by the two approaches. To that end one could characterize the non-stationary noise (plus other instrumental contributions and spectroscopic uncertainties if relevant) and its corresponding error in delta tau and plot it as error envelope in Fig. 5.

Specific comments:

On page 8, line 26: Why do you need steps 1 and 2? To be close to the local minimum of your error function?

P6, l 25: It is not clear to me how you stabilized the instrument baseline. As far as I understand you try to keep the ON laser wavelength at maximum absorption. But the rest is unclear. What's the role of the thermal controller? Did you lock the OFF wavelength to the ON wavelength via the differential optical depth?

---

## Author Comment (AC1) · 31 Jul 2019

Dear Reviewer,

Thank your for your feedback. The following contains your original comments and the author/co-authors response to each comment and proposed additions to the paper.

Sincerely, Scott Zaccheo

The paper addresses the non-trivial comparison between CO2 concentrations from in situ sensors with those from remote sensors, which measure path-integrated (path averaged) CO2 concentrations. This work represents a very crucial step towards improved quantification of sources and sinks of greenhouse gas emissions as it may provide spatially integrated concentrations important for flux retrieval. This is especially important as urban centres become significant producers of greenhouse gases. Regional emission modeling frameworks will benefit from this work. C1 AMTD Interactive comment Printer-friendly version Discussion paper It is a clearly written paper and the limitations of the approach are discussed in the end so I recommend publication. I just have a few comments, see below.

General comment

In essence, if I'm not wrong, you calibrate the differential optical depth of your open path differential absorption spectrometer with point optical depths from in-situ sensors. It works and the result is impressive but the method is pragmatic as you state yourselves in the manuscript already. This must assume that the in-situ sensors are accurate, which is reasonable, but only near them, which is the dilemma of comparing path averaged CO2 concentrations with point concentrations. Don't you risk adding a new bias by this method? In that context it would be interesting to know how much of the difference between GreenLite and the in-situ values are caused by instrument bias and random error and how much are actually "real", that is, physical differences in CO2 concentrations due to the different volumes probed by the two approaches. To that end one could characterize the non-stationary noise (plus other instrumental contributions and spectroscopic uncertainties if relevant) and its corresponding error in delta tau and plot it as error envelope in Fig. 5.

Response: The reviewer has very succinctly stated the problem that the authors and a number of other collaborators have contemplated, discussed and argued over for a significant period of time. In hindsight, we would do several things to address the question of what is the "true", real differences/instrument errors/etc between the in

situ and column measurements given the opportunity in the future. A real possibility exists, albeit small, that the uncorrected column observations are unbiased to start out with. However, trying to realistically characterize the lump error contribution of the measurement approach, given the bias correction, in any unbiased fashion at this juncture from these data is problematic due to several factors that include a lack of independent and controlled observation either on site and/or pre/post deployment.

Specific comments

On page 8, line 26: Why do you need steps 1 and 2? To be close to the local minimum of your error function?

Response: Step #1 (Randomly select a sparse set of samples from the entire set of all chords on a transceiver-by- transceiver basis) and #2 (Use a nominal model spectra computed based on the corresponding average temperature, relative humidity, and atmospheric to locate the wavelength of maximum absorption and assign this value as the optimized on-line wavelength for the transceiver sample) were included as part of the process for a number of reasons. Over the course of a year we collected over 6x106 observations that were deemed of sufficient quality to include in this study. Applying an iterative search algorithm to identify an optimal off-line fit for each 10 sec sample was simply not feasible given the available internal resources, and deemed to be unwarranted given that we were looking to correct for a slowly varying bias term that was on the order of days. The final estimated off-line position for any given retrieval was constructed as the median value over a 4 day window, making it most likely unnecessary to compute all values in any given 4 day window to construct a reasonable value. For these reasons we devised a random selection method that took into account the desire to incorporate measurements at all chord positions over as wide a range of times during the day as possible in an unbiased fashion. The application of step #2 was designed to compute a reasonable on-line wavelength that was known to be set at the wavelength of observed peak absorption. We recognize that the continuous calibrations of the instrument wavelengths were sub optimal in this design. However we made a best effort to compensate for this by remotely tuning the on-line to a wavelength of maximum absorption for a given chord within a narrow window, and adjusting the off-line thermal control, which acts as a mechanism for fine tuning each wavelength, by the same amount. This short coming has been addressed in our current design (see discussion below). Given our wavelength calibration method, and the fact that we could correct for only one free parameter, we computed the expected wavelength of maximum absorption given the atmospheric state and a nominal column concentration, assigned the on-line wavelength to this value and found the off-line wavelength that provided the best fit given the corresponding in situ measurement. While not perfect it did provide what seems to be a robust and automated mechanism for computing off-line wavelengths that could be averaged over long periods of time (days) to provide physically realistic values, within $\pm 10$ pm from measured off-line positions, and that achieved the desired goals.

In order to clarify the process, we propose to add the following to the paragraph in question.

"A random sampling approach was selected in Step 1 to reduce the computational burden associated with locating an optimal off-line at each sample, while preserving the associated median statistics over the multi-day window of interest, and the optimization of the off-line was chosen over that of the on-line based on the fact that the on-line was adjusted on a regular basis to match the observed maximum absorption."

P6, l 25: It is not clear to me how you stabilized the instrument baseline. As far as I understand you try to keep the ON laser wavelength at maximum absorption. But the rest is unclear. What's the role of the thermal controller? Did you lock the OFF wavelength to the ON wavelength via the differential optical depth?

Response: The thermal controller serves as the fine adjustment mechanism for the on-line and off-line laser wavelengths. We started the project using a high accuracy absolute wavelength reference to measure both the online and the offline wavelengths.

When we noticed they appeared to be drifting we implemented the approach described in the paper since we were operating remotely and did not have access to the absolute wavelength measurement. The online was able to use the maximum absorption as a pseudo-wavelength reference, but without an equivalent reference for the offline we operated under the assumption that the equivalent adjustments needed to bring the online to the maximum absorption were also what was needed to correct the drift for the offline, given the lasers were the same part numbers from the same manufacturer. Testing after the Paris demonstration found that although the online and offline wavelength control behave very similarly there are slight differences in the response which would have resulted in slightly incorrect corrections of the offline drift over the course of the experiment. There is no clear method to retrospectively quantify these errors in terms of absolute wavelength, and this has been addressed in our current implementation by adding a gas cell and fully characterizing the response of the individual lasers.

---

## Referee Comment (RC2) · Anonymous Referee #2 · 2 Aug 2019

This paper describes bias correction method for long-path absorption CO2 measurements. The subject is suitable for AMT, and the paper is well written.

Specific Comments p.11 I.4: "on-line position" -> "off-line position"

Table 1: The table heading "Average Wavelength Offset (GreenLITETM-in situ)" is not very correct expression.

It would be better to add a brief description on GreenLITETM hardware system including key words such as "DFB laser diode" and "Semiconductor optical amplifier".

Otherwise it is not possible to figure out the hardware system without seeing Dobler et al. 2017.

p.3, I. 24: The following description seems not correct. "the common mode terms cancel out for the IMCW approach but would be independent for the better-known pulsed method." Common mode terms are cancel out with pulsed methods too.

The observation site names are sometimes difficult to follow. Is "CTI tower" the same as "the roof of the lower of the two Montparnasse building"? Jussieu (p.8. I. 33) should be QUA. It would be better to indicate the GreenLITE site names in Fig. 1.

---

## Author Comment (AC2) · 16 Aug 2019

The authors would like to thank Reviewer #2 for their very insightful comments. Each comment (indicated by C:) is listed below along with individual responses denoted by R:

**Specific Comments**

C: p.11 I.4: "on-line position" -> "off-line position"

R: The reviewer is quite correct. It should be off instead of on, and will be corrected in the final draft. Thank you.

C: Table 1: The table heading "Average Wavelength Offset (GreenLITETM-in situ)" is not very correct expression.

R: The authors agree with the reviewer. The label will be changed to "Average Wavelength Offset (Nominal - Postprocessed)"

C: It would be better to add a brief description on GreenLITETM hardware system including key words such as "DFB laser diode" and "Semiconductor optical amplifier". Otherwise it is not possible to figure out the hardware system without seeing Dobler et al. 2017.

R: The focus of this paper is to describe a method of correcting a difference between a long open-path differential absorption measurement and a point concentration measurement and not to fully describe the hardware which has been the subject of previous publication. We provided an overview of the measurement method that describes how the measurements are being made, which seems adequate for the purpose of this paper.

C: p.3, l. 24: The following description seems not correct. "the common mode terms cancel out for the IMCW approach but would be independent for the better-known pulsed method." Common mode terms are cancel out with pulsed methods too.

R: There are a number of terms that are not common mode for pulsed implementations that are for the IMCW approach. The key advantage of the IMCW approach in this regard is the simultaneity of the online and offline signals both in transmission and reception. For example, a pulsed method will see a slightly different ground reflectivity for the online versus the offline due to the time delay between pulses that is not present for the IMCW approach where the same ground spot is simultaneously sampled. Other examples include: 1) atmospheric scintillation, 2) receiver electronics and detector

noise, and 3) optical amplifier noise, each of these are independent terms for the two time-delayed channels of a pulsed system but are common for the IMCW approach. We propose to change the text to more accurately state:

"Since the differential transmission is determined through a ratio of the transmitted and received signals at the different wavelengths, there are a number of terms that are common mode for the IMCW approach due to simultaneous transmission of the online and offline wavelengths that cancel out but would remain independent for the better-known pulsed method."

**Versus original text**

"The design is such that several noise sources are now common mode due to the simultaneity. Since the differential transmission is determined through a ratio of the transmitted and received signals at the different wavelengths, the common mode terms cancel out for the IMCW approach but would be independent for the better-known pulsed method."

C: The observation site names are sometimes difficult to follow. Is "CTI tower" the same as "the roof of the lower of the two Montparnasse building"? Jussieu (p.8. I. 33) should be QUA. It would be better to indicate the GreenLITE site names in Fig. 1.

R: The authors agree with the review. All the references to CIT, Jussieu, CDS and QUA have been review and changed to the following consistent set of identifiers: CIT has been defined as the "Tour CIT Montparnasse building (CIT)", JUS/Jussieu as the "the Jussieu tower at UPMC (JUS), CDS as the "Cite des Sciences et de l'Industrie (CDS)" and QUA as the "QUALAIR

**C3**

---

## Author Comment (AC3) · 27 Aug 2019

The response to these comments are provided in AC2 that was inadvertently not linked to RC2. the link is https://editor.copernicus.org/index.php/amt-2019-199-AC2.pdf?_mdl=msover_md&_jrl=400&_lcm=oc108lcm109w&_acm=get_comm_file&_ms=76